# Decelerated Epigenetic Aging in Long Livers

**DOI:** 10.3390/ijms242316867

**Published:** 2023-11-28

**Authors:** Zulfiya G. Guvatova, Anastasiya A. Kobelyatskaya, Elena A. Pudova, Irina V. Tarasova, Anna V. Kudryavtseva, Olga N. Tkacheva, Irina D. Strazhesko, Alexey A. Moskalev

**Affiliations:** 1Russian Clinical Research Center for Gerontology, Pirogov Russian National Research Medical University, Ministry of Healthcare of the Russian Federation, Moscow 129226, Russia; irish.ff@mail.ru (I.V.T.); tkacheva@rgnkc.ru (O.N.T.); istrazhesko@gmail.com (I.D.S.); 2Engelhardt Institute of Molecular Biology, Russian Academy of Sciences, Moscow 119991, Russiapudova_elena@inbox.ru (E.A.P.); rhizamoeba@mail.ru (A.V.K.)

**Keywords:** aging, epigenetic clock, DNA methylation, long livers, pyrosequencing

## Abstract

Epigenetic aging is a hot topic in the field of aging research. The present study estimated epigenetic age in long-lived individuals, who are currently actively being studied worldwide as an example of successful aging due to their longevity. We used Bekaert’s blood-based age prediction model to estimate the epigenetic age of 50 conditionally “healthy” and 45 frail long-livers over 90 years old. Frailty assessment in long-livers was conducted using the Frailty Index. The control group was composed of 32 healthy individuals aged 20–60 years. The DNA methylation status of 4 CpG sites (*ASPA* CpG1, *PDE4C* CpG1, *ELOVL2* CpG6, and *EDARADD* CpG1) included in the epigenetic clock was assessed through pyrosequencing. According to the model calculations, the epigenetic age of long-livers was significantly lower than their chronological age (on average by 21 years) compared with data from the group of people aged 20 to 60 years. This suggests a slowing of epigenetic and potentially biological aging in long livers. At the same time, the obtained results showed no statistically significant differences in delta age (difference between the predicted and chronological age) between “healthy” long livers and long livers with frailty. We also failed to detect sex differences in epigenetic age either in the group of long livers or in the control group. It is possible that the predictive power of epigenetic clocks based on a small number of CpG sites is insufficient to detect such differences. Nevertheless, this study underscores the need for further research on the epigenetic status of centenarians to gain a deeper understanding of the factors contributing to delayed aging in this population.

## 1. Introduction

Aging is a multifactorial biological process accompanied by alterations in molecular, cellular, and physiological functions in most living organisms, resulting in a progressive decline in overall fitness and, eventually, death. Accumulated evidence indicates that age-associated alterations in DNA methylation patterns, along with other epigenetic modifications such as histone modifications, chromatin remodeling, or non-coding RNAs, play a crucial role in aging and the development of age-related diseases [1]. To date, changes in DNA methylation levels of specific CpG sites are widely used as aging biomarkers to estimate an individual’s age in prediction models, currently referred to as the epigenetic clock [2]. There are many variants of epigenetic clocks that quantify biological age across tissues in whole blood, skin, muscle, and cell cultures [3]. These models are based on machine learning and deep learning algorithms and are able to predict not only age but also mortality and the occurrence of aging-related diseases, such as cardiovascular diseases, diabetes, cancers, and Alzheimer’s disease [4]. For example, the acceleration of epigenetic age was associated with an increased risk of developing breast cancer [5]. Epigenetic age was higher in patients with Alzheimer’s disease [6], COVID-19 [7,8], Werner syndrome [9], and obesity [10]. Chronic alcohol consumption also accelerates epigenetic age by an average of 2.2 years [11]. Moreover, some anti-aging interventions have been shown to reverse the epigenetic clock while alleviating the phenotypic manifestations of aging [12,13].

However, to date, there are no standardized or widely accepted aging biomarkers and prognostic models. Before considering the use of epigenetic clocks in routine clinical practice, it is necessary not only to improve the accuracy of prediction and reduce the cost of analysis but also to examine the existing clocks in independent studies and validate them on wide age ranges and new populations.

In this context, long-lived individuals, as models of successful aging, are a population of great interest. Long livers are a group of people aged 90 years or above who exhibit better cognitive and functional status and tend to have a lower risk of all-cause mortality, as well as age-associated diseases, such as cancer and cardiovascular diseases. 

The present study estimated epigenetic age in Russian long livers using the epigenetic clock based on a small number of CpG sites. In addition, the study addressed the question of whether the epigenetic age would be associated with frailty in long livers. As an age prediction model, the Bekaert clock was selected, which has shown accurate age predictions for blood samples from people aged 0 to 91 years [14]. Among the CpG sites located in age-associated genes, the authors selected four sites (*ASPA* CpG1, *PDE4C* CpG1, *ELOVL2* CpG6, and *EDARADD* CpG1) that turned out to be the most informative for predicting age. In addition, this model, along with others, was later successfully used by Daunay et al. to estimate the epigenetic age of centenarians in the French population [15].

## 2. Results

The aim of the study was to estimate the epigenetic age of long livers and individuals from the general population aged from 20 to 60 years, using the Bekaert age prediction model [14,15,16]. The DNA methylation levels of CpG sites and the predicted ages of individuals can be found in Appendix A. In the group of individuals aged 20–60 years, the results of the correlation analysis showed a statistically significant relationship between the DNA methylation of each CpG included in the model and the chronological age (2.87 × 10^−11^ ≤ *p*-value ≤ 2.23 × 10^−6^; Figure 1). *ELOVL2* CpG_6_ and *PDE4C* CpG_1_ presented a strong positive correlation (r = 0.88, r = 0.81), while the *ASPA* CpG_1_ and *EDARADD* CpG_1_ showed strong negative correlations (r = −0.72, r = −0.79). When analyzing the entire sample, a weakening of the relationship between *ELOVL2* CpG_6_ methylation and age was observed (r = 0.23, *p*-value = 0.009), which may be attributed to the observed dispersion of values in the group of long livers, as can be seen in Figure 1. For *PDE4C* CpG_1_, *ASPA* CpG_1_, and *EDARADD* CpG_1_, the Spearman correlation coefficients were 0.64 (*p*-value = 2.89 × 10^−16^), −0.64 (*p*-value = 3.65 × 10^−16^), and −0.59 (*p*-value = 3.84 × 10^−13^), respectively.

According to the calculations, the epigenetic age of long livers was much lower than their chronological age, with an average difference of 21 years. In the control group of individuals aged 20 to 60 years, the variance between the predicted age and chronological age was significantly lower. As can be seen in Figure 2a, in the control group, there were both underestimated and overestimated predictions (−2 and 4.8 years on average). Subsequently, the delta age, representing the difference between the predicted age and chronological age, was calculated for each person. One-way analysis of variance (ANOVA) showed that the difference in delta age between the groups was significant (F = 93.20, *p*-value = 1.96 × 10^−25^). Post hoc analyses using the Scheffe post hoc criterion for significance indicated that delta age was significantly different in the control group versus frail long livers (*p*-value = 4.36 × 10^−22^) and “healthy” long livers (*p*-value = 7.808100 × 10^−22^) (Figure 2b). There was also an interest in determining whether epigenetic age would differ among long livers with and without frailty. However, the obtained results showed no statistically significant differences in delta ages between these groups (*p*-value = 0.92) (Figure 2b).

The prediction accuracy of the model was evaluated using the median absolute deviation (MAD) between the predicted and chronological ages. MAD values were calculated for the control group, frail long livers, and “healthy” long livers separately. As expected, for the group of individuals aged 20–60 years, the smallest MAD was obtained (MAD = 4.3), and for the frail long livers and “healthy” long livers, MAD values were equal to 8.1 and 6.6, respectively.

Several previous studies have reported sex differences in age-associated methylation. However, the current study did not find significant differences in delta age between the sexes in all groups.

## 3. Discussion

The present research assessed epigenetic age in Russian long-lived men and women, as well as in individuals from the general population aged from 20 to 60 years, using the Bekaert age prediction model based on four CpG sites. According to our results, the epigenetic age of long-lived individuals was significantly lower than their chronological age, on average by 21 years, while among individuals aged from 20 to 60 years, the delta age values were much smaller. These results suggest a deceleration of the aging rate, specifically in long-lived people, leading to a reduction in epigenetic age. Lower epigenetic age compared to the chronological age in people older than 90 has also been observed in other studies using epigenetic clocks based on both a small number of CpG sites and a large number of CpG sites [15,16,17,18]. For example, using four epigenetic clocks based on a small number of CpG sites, Daunay et al. showed identical results to ours in French centenarians and semi-supercentenarians [15].

Regarding sex differences in epigenetic age, comparisons between men and women did not reveal any significant differences in all groups. This lack of distinction can be explained, in part, by the limited number of CpG sites in Bekaert’s clock. Studies that have identified sex-specific effects on epigenetic age typically use clocks with a large number of CpG sites. Some previous research has shown that women tend to have a slightly younger epigenetic age than men, suggesting that biological aging may progress at a slower rate in women [19,20]. Notably, despite having worse health at the end of life, females tend to live longer than men [21]. Moreover, several studies have shown that women exhibit more frailty than men across all age groups, including long livers [22]. These findings emphasize the need for in-depth studies to understand better the underlying factors contributing to sex differences in the aging process.

Another goal of this study was to compare the epigenetic age of long livers with frailty and «successfully aging» long livers. Environmental and lifestyle factors can have differential effects on the clinical profile and epigenetic status over the course of a person’s life [23]. Consequently, as a result, long livers constitute a highly heterogeneous population, encompassing individuals in both good and very poor health statuses, as demonstrated by the varying degrees of frailty [24]. Several previous studies have explored the association between epigenetically determined age and frailty, and these studies have confirmed a strong link between prevalent frailty and epigenetic age [25]. However, in the current study, no statistically significant differences in delta age (the difference between the predicted and chronological age) between the groups with and without frailty were observed. It is possible that Bekaert’s clock cannot detect such differences due to the absence of specific CpG sites. It has been shown that many existing epigenetic clocks are weakly correlated with each other, and they can capture different aging pathways and hallmarks, such as cellular senescence, mitochondrial dysfunction, inflammaging, etc. [26]. A recent longitudinal study of the association between frailty and longevity has found that clocks trained on phenotypic markers and mortality were better predictors of changes in frailty when compared with clocks solely trained on chronological age [25]. For example, GrimAge takes into account changes in levels of the C-reactive protein, which is significantly associated with changes in frailty [27,28]. 

On the other hand, it is important to consider that the majority of currently published epigenetic age prediction models, including most popular clocks such as GrimAge, Hannum, and Horvath, were trained on data that included results of the DNA methylation of people younger than 80 years, with a smaller proportion of people of old age in the training set [29,30,31,32]. Therefore, these epigenetic clocks can be less accurate in predicting the age of long livers. As of the time of this writing, only one study has presented an epigenetic clock trained on centenarian samples. The model, called “Centenarian clock”, predicts the age of centenarians with a lower median absolute error (MAE = 1.8 years) than other clocks, which often tend to underestimate the age of older people [33]. Given the features of long livers aging, the development of clocks for people older than 90 years holds promise for the study of the phenomenon of “successful aging”.

Taken together, the obtained results demonstrate that Russian long livers have a significantly younger epigenetic age compared to their chronological age, implying that they experience a delay in epigenetic aging. Although research involving long-lived individuals, especially those with frailty, poses its own set of challenges and recruitment difficulties for experimental studies, there is a clear need for further research on this population. Such research is essential for gaining a better understanding of aging in general and for developing effective personalized medicine approaches to protect against age-related diseases.

## 4. Materials and Methods

### 4.1. Study Participants

This study used data obtained at the Russian Clinical Research Center of Gerontology during a study aimed at identifying factors associated with longevity in the Russian population. The participants aged 89–107 years provided questionnaire data through in-home interviews and blood samples. Subsequently, 52 women and 43 men were selected for DNA methylation analysis, of which 25 women and 20 men exhibited frailty. Frailty assessment in long livers, was conducted using the Frailty Index approach, which considers health/well-being disorders (referred to as deficits) [34]. The following deficit variables were selected: help bathing, help dressing, help getting in/out of a chair, help walking around the house, help eating, help grooming, help using the toilet, help up/down stairs, help shopping, help with housework, help with meal preparations, help taking medication, help with finances, have lost more than 5 kg in last year, self-rating of health, walking outside, feel everything is an effort, feel depressed, arterial hypertension, cardiovascular diseases, chronic heart failure, stroke, cancer, diabetes, arthritis, chronic lung disease, MMSE, BMI, and grip strength [35]. The Frailty Index was calculated as the proportion of deficits present relative to the total sum of considered deficits, ranging from 0 to 1 (increasing values represent worse health). Frailty was defined as being absent for any participant with FI < 0.35. Two groups were formed: one comprising frail long livers and another consisting of conditionally “healthy” long livers. The control group was composed of 32 healthy individuals aged 20–60 years. As a result, the entire sample consisted of 127 individuals.

### 4.2. DNA Extraction and Quantification

Total DNA was isolated and purified using a DNA-sorb-B kit (AmpliSens, Moscow, Russia) following the manufacturer’s protocol. The DNA concentration was determined using a Qubit 2.0 Fluorometer (Thermo Fisher Scientific, Waltham, MA, USA).

### 4.3. DNA Methylation Analysis

A sample of 500 ng of blood-extracted DNA was bisulfite-treated using an EZ DNA Methylation-Gold Kit (Zymo Research, Irvine, CA, USA). A sample of 20 ng of bisulfite-treated DNA was used as a template for each PCR using the PyroMark PCR kit (Qiagen, Hilden, Germany). Four primer pairs were used: *ASPA, EDARADD, ELOVL2*, and *PDE4C*. Primers for PCR and pyrosequencing were taken from [15] and are presented in Appendix A. Description of the 4 genes and genomic coordinates of CpG sites are presented in Appendix A.

The following conditions were used for PCR amplification: initial denaturation at 95 °C for 15 min followed by 50 cycles of denaturing at 95 °C for 30 s, annealing at 52 °C for the *ASPA* gene, 56 °C for *EDDARAD*, 60 °C for *ELOVL2*, 53 °C for the *PDE4C* gene for 30 s and extension at 72 °C for 30 s, followed by the final extension at 72 °C for 10 min. After amplification, the PCR product was checked by agarose gel analysis. Then, 10 μL of the PCR product were used for pyrosequencing on the PyroMark Q48 Autoprep instrument (Qiagen, Hilden, Germany) using PyroMark Q48 Advanced CpG Reagents (Qiagen) according to the manufacturer’s protocol. Data were generated and analyzed with the PyroMark Q48 Autoprep Software (ver. 4.3.3), allowing analysis in CpG mode (Qiagen). The methylation percentage of individual CpG sites that passed built-in quality control for bisulfite treatment was used for further analysis.

The Bekaert clock was used as a blood-based age prediction model, which showed accurate age predictions for blood samples from people aged 0 to 91 years [14]. The Bekaert clock is based on the methylation level of 4 CpG sites located in 4 genes: *ASPA* (CpG1), *PDE4C* (CpG1), *ELOVL2* (CpG6), and *EDARADD* (CpG1) and has the following formula:26.444119 − 0.201902 × *ASPA* (CpG1) − 0.239205 × *EDARADD* (CpG1) + 0.0063745 × *ELOVL2* (CpG6) ^2^ + 0.6352654 × *PDE4C* (CpG1).

### 4.4. Statistical Analysis

Visualization and statistical analysis were performed using Python (ver. 3.6) in Jupyter Notebook (ver. 6.4.0). Correlation analyses were performed by calculating the Spearman’s rank coefficient of correlation. The delta age, representing the differences between the predicted age and chronological age, was calculated for each subject. Statistical significance was evaluated using one-way ANOVA with Scheffe’s method. The results were considered significant at *p*-value < 0.05.

## Figures and Tables

**Figure 1 ijms-24-16867-f001:**
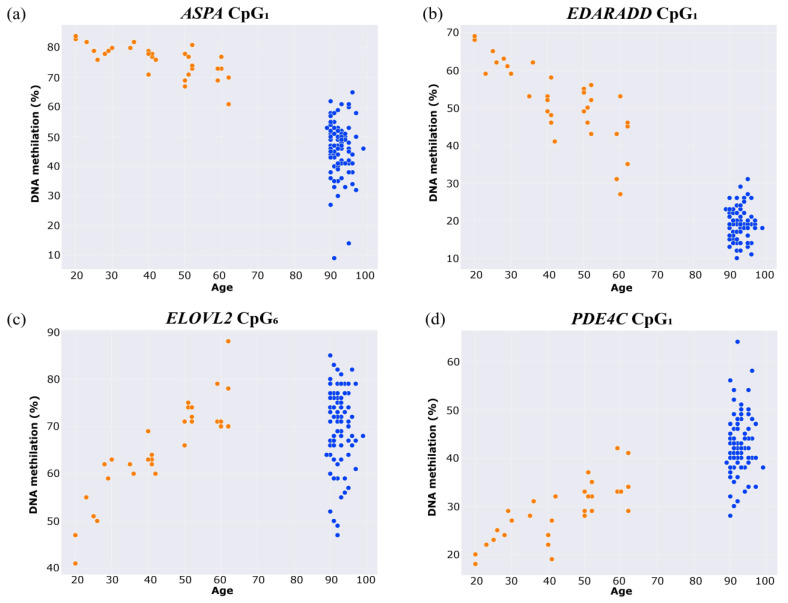
Scatter plots of the correlation between chronological age and DNA methylation of (**a**) *ASPA* CpG_1_, (**b**) *EDARADD* CpG_1_, (**c**) *ELOVL2* CpG_6_, and (**d**) *PDE4C* CpG_1_. The control group is marked in orange, long livers—blue.

**Figure 2 ijms-24-16867-f002:**
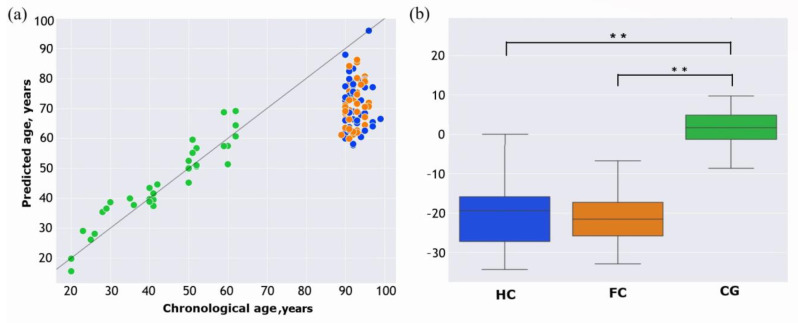
Comparison of the predicted and chronological age. (**a**) Scatterplot of the predicted age against chronological age for each group: CG—control group (green), HC—“healthy” long livers (blue), FC—frail long livers (orange). The line corresponds to the case when the predicted age is the same as the chronological age. (**b**) Boxplots of the delta age (difference between the predicted and chronological age) according to each group. ** *p*-value < 0.001, one-way ANOVA.

## Data Availability

Data is contained within the article or Appendix A.

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
