# Peer review of "Decelerated Epigenetic Aging in Long Livers"

_ijms, 2023, doi:10.3390/ijms242316867_

Round 1
Reviewer 1 Report
Comments and Suggestions for Authors
The manuscript presents an interesting study on DNA methylation analysis of four genes in the context of aging. Overall, the study is well-conducted and adds valuable insights into the epigenetic regulation of these genes. However, there are several points that need to be addressed to strengthen the paper's quality.
1. Show the gene structure and CpG site locations of the 4 genes. The manuscript would benefit from a more detailed presentation of the gene structures and the specific CpG site locations analyzed. This information is crucial for readers to understand the context of DNA methylation analysis. Please provide gene structure diagrams and clearly indicate the locations of the CpG sites.
2. Please provide background and rationale for the 4 genes selected. The authors need to provide a more comprehensive background on the four selected genes. Explain why these specific genes were chosen for the study and what is known about their relevance to aging. Additionally, provide a rationale for the selection of the specific CpG sites within these genes.
3. It is essential to include both positive and negative control genes for the DNA methylation analysis of the four genes. These controls will help validate the accuracy and specificity of the methylation measurements.
4.To provide a more holistic view of DNA methylation patterns within the selected genes, it is recommended that the authors include data on DNA methylation levels of other CpG sites within these genes. This will help readers understand the overall methylation landscape of these genes and how the specific CpGs analyzed relate to the broader context.
5.In Figure 2, the authors have used the Mann-Whitney U test for statistical analysis. It is crucial to explain why this test was chosen over other statistical tests, such as one-way ANOVA. Provide a rationale for the choice of the Mann-Whitney U test, especially in the context of the data presented in Figure 2.
6.The use of Pearson correlation for analyzing the relationship between variables needs further justification. Given the potential non-linear relationships in biological data, the authors should explain why Pearson correlation was preferred over non-linear models.
In summary, this study offers valuable insights into DNA methylation of four genes in aging. Addressing the points mentioned above will enhance the quality and clarity of the manuscript, making it more accessible to the scientific community.
Comments on the Quality of English LanguageThe manuscript's writing needs improvement for clarity and coherence. Consider revising the text to ensure that the research findings are presented in a logical and organized manner. Additionally, ensure that the language and terminology are clear and consistent throughout the paper.
Author Response
1. Description of the 4 genes and genomic coordinates of CpG sites has been added to Supplementary Table 2. Although we agree that this is an important сonsideration, we do not consider it appropriate to include this information as well as gene structure diagrams in the main text because it is beyond the scope of the Communication.
2. We have reworked the Introduction section for clarity. The selection of specific genes and specific CpG sites is made during development of an epigenetic age prediction models. We did not develop a new model in our work. We chose between available models that were applicable to blood samples and based on a small number of CpG sites.
We tried to clarify the point: «The present study estimated epigenetic age in long-livers using the epigenetic clock based on a small number of CpG sites. In addition, the study addressed the question of whether the epigenetic age would be associated with frailty in long-livers. As an age prediction model, the Bekaert clock, was selected, which showed accurate age predictions for blood samples from people aged 0 to 91 years [14]. Among the CpG sites located in ageassociated genes the authors selected 4 sites (ASPA CpG1, PDE4C CpG1, ELOVL2 CpG6, EDARADD CpG1) that turned out to be the most informative for predicting age. In addition, this model, along with others, was later successfully used by Daunay et al. to estimate the epigenetic age of centenarians in the French population [15]. »
3. We agree that this is can be potential limitation of the study. And at the same time, data generated with the methylation analysis software (detection and quantification of CpG sites) on PyroMark Q48 contains unique features that act as quality control for complete bisulfite conversion of DNA. Built-in controls for the bisulfite treatment eliminate manual estimation of non-converted DNA levels and prevent false-positive methylation detection, thereby ensuring the reliability of results.
4. We think this is an excellent suggestion. Data on DNA methylation levels of other CpG sites have been included in Supplementary Table 1.
5. Thank you for pointing this out. Indeed, one-way ANOVA is more appropriate for these data. We reanalyzed the data and have added corrections to the results and statistical analysis section. The conclusions have not changed.
6. This observation is correct. We have calculated the Spearman correlation coefficient and added corrections to the results. The conclusions have not changed.

Reviewer 2 Report
Comments and Suggestions for Authors
The authors should be commended for the valuable study, which holds great potential for assisting the elderly population and delaying the aging process by identifying key biomarkers.
Although, there have been similar studies in the past, the authors should highlight the novelty of this study (For example: Daunay et al. have shown identical results in terms of "epigenetic age of centenarians in the French population).
There are a few points which should be taken care as described below:
Major comments:
1. Abstract is not well written, methodology should also be involved in abstract including number of samples.
2. The total of samples is missing, there were 52+44 centenarians but no information on 20-60 group.
Minor comments:
1. Line 52: "Centenarians are a group of people aged 90 years or above" but Centenarians are by definition above 100 years.
2. Line 84: Please use appropriate scientific language instead of "not so great".
3. Line 87: "of" is missing after age.
4. Line 89: There is only one p value is provided, is that correct?
5. Line 96: What is smallest MAD? MAD is just one value. Are you comparing with centenarians? If yes, then mention it.
Apart from these there are some minor grammatical errors like use of singular or plural, use of proper verb form and use of comma etc. For example:
Line 71: please check if "sample" is correct or a plural form is needed.
Line 75: were instead of was
Line 142: inflammaging
Line 158: phenomena as
Comments on the Quality of English LanguageThere are also minor grammatical/spelling errors throughout manuscript, which I have not mentioned exclusively.
Author Response
Thank you for the review! We appreciate the time and effort that you have dedicated to providing your valuable feedback on our manuscript. We have studied comments carefully and have made corrections as marked in the revised manuscript, which we sincerely hope will meet with your approval.
Major comments:
- The Abstract section has been reworked
- We agree with Reviewer. Information about the number of participants in the control group and the total number of participants has been added to the “4.1. Study participants” section.
Minor comments:
- Thank you for pointing this out. In the revised version of the manuscript “centenarians” was replaced by “long-livers”
- This sentence has been rephrased.
- It’s not entirely clear where exactly "of" is missing. In any case, the manuscript has been proofread. We have attached the certificate.
- Thank you for this comment. Indeed, there should be 2 p-values. The second p-value have been added.
- Evaluation of mean absolute deviation (MAD) between chronological and predicted ages in control group, frail long-livers and «healthy» long-livers was performed separately.
The sentence has been rephrased for clarity. - - 9. Correction has been made

Round 2
Reviewer 1 Report
Comments and Suggestions for Authors
The authors have addressed most of my concerns, and the manuscript is at better shape towards publication. It would be very helpful if the authors can present the control CpG sites data together with the significant one in the figures.
Comments on the Quality of English LanguageThe quality of the writing could benefit from enhancement.
Author Response
Dear Reviewer,
Thank you for the comments!
Some corrections have been made in the «4.3. DNA methylation analysis» section (Line 219) for more clarity. For performing quality control for complete bisulfite
conversion of DNA PyroMark Software doesn’t use control CpG sites. Built-in quality control for bisulfite treatment is based on the fact that all unmethylated C’s not followed by a G are completely converted to a T after bisulfite treatment and PCR. This acts as a useful quality control for full conversion of unmethylated C residues during bisulfite treatment and PCR. In addition, the software provides a reference sequence pattern, delivering information about the location of expected peaks and expected peak heights. Post-run, the software performs a quality assessment of individual sites as well as the sequence context based on this information.
Previously, a number of studies (10.1186/s12915-020-00807-2, 10.7554/eLife.37462, 10.3389/fgene.2021.772298) have also successfully used a similar approach (genomic DNA " bisulfite conversion " PCR " Pyrosequencing on PyroMark System " analysis with PyroMark Software, allowing analysis in CpG mode) to estimate epigenetic age.
The manuscript has been proofread by a native English speaker. We have attached the certificate.
